# Canine Caregivers: Paradoxical Challenges and Rewards

**DOI:** 10.3390/ani12091074

**Published:** 2022-04-21

**Authors:** Lori R. Kogan, Jean E. Wallace, Peter W. Hellyer, Eloise C. J. Carr

**Affiliations:** 1College of Veterinary Medicine and Biomedical Sciences, Colorado State University, Fort Collins, CO 80523, USA; peter.hellyer@colostate.edu; 2Department of Sociology, University of Calgary, 2500 University Drive NW, Calgary, AB T2N 1N4, Canada; jwallace@ucalgary.ca; 3Emeritus, Faculty of Nursing, University of Calgary, 2500 University Drive NW, Calgary, AB T2N 1N4, Canada; ecarr@ucalgary.ca

**Keywords:** canine, aging, senior, caregiver, human animal bond

## Abstract

**Simple Summary:**

People with aging or ill family members often fill the role of caretaker. Companion dogs are often viewed as family members and because they age more rapidly than people and have shorter lifespans, having a dog often includes caring for it during its senior years. Caring for an elderly dog can be physically and emotionally challenging, yet we know little about how caring for an aging dog impacts guardians’ lives. This study was designed to better understand dog guardians’ experiences and perceptions related to caring for their aging dog. We asked dog guardians to complete an online anonymous survey, resulting in a sample size of 284 participants. We found that the impact on guardians when caring for an aging dog appear to share many similarities with caregivers of human family members. Our results suggest that, for many guardians, caring for an aging dog is a complex dynamic with both positive and negative factors that offers an opportunity to deepen the human-animal bond and create positive, rewarding experiences and memories.

**Abstract:**

Companion dogs are increasingly popular, 38.4% of households in the United States include at least one dog. There are numerous benefits to sharing one’s home with a dog, but because they age more rapidly than people and have shorter lifespans, acquiring a dog often includes caring for it during its senior years. Caring for an elderly dog can be physically and emotionally challenging, yet the impact on guardians’ lives when caring for an aging dog has received minimal scientific attention. This study was designed to better understand dog guardians’ experiences and perceptions related to caring for their aging dog. Utilizing an exploratory mixed methods design, this study asked dog guardians to complete an online anonymous survey. From a total of 284 participants, we found that the impact on guardians when caring for an aging dog appears to share many similarities with caregivers of human family members. Our quantitative and qualitative results suggest that, for many guardians, caring for an aging dog is a complex dynamic with both positive and negative factors that offers an opportunity to deepen the human-animal bond and create positive, rewarding experiences and memories.

## 1. Introduction

An aging population, as well as an increased number of people living with chronic disease, has increased the number of family members acting as caregivers [1]. Similarly, with advances in veterinary medicine, caring for aging pets is also becoming a reality for many families. This paper focuses on caring for an aging dog, using the rich collection of research pertaining to human caretaking as an important theoretical foundation.

### 1.1. Caregivers and Family Members

People with aging or ill family members often fill the role of caretaker; providing unpaid, ongoing assistance with activities of daily living [2]. Yet, these family members are often unprepared for this challenging new role, many times creating a wide range of unmet needs for themselves and those they care for [3]. In addition to the physical and emotional demands associated with caregiving, many caretakers find the people they provide care for, often times the very same people they have received emotional support from in the past, are no longer able to offer support [4]. Because of this, informal family caregiving is often described as a chronically stressful experience [5] with the resultant caregiver burden defined as the “multidimensional biopsychosocial reaction resulting from an imbalance of care demands relative to caregivers’ personal time, social roles, physical and emotional states, financial resources, and formal care resources given the other multiple roles they fulfill” [6].

Several studies have found that caregiving, a role that often spans many years, can affect caretakers’ physical and emotional health as well as their quality of life [7,8,9,10]. This includes increased levels of anxiety and depression, which can negatively impact both the caretaker and the recipient [7,11,12,13,14,15,16,17,18]. It is, therefore, not surprising that Schulz [19] suggests that caring for an elderly individual with disabilities is burdensome and stressful and that family caregivers perform this service at considerable cost to themselves.

Yet, these negative consequences do not appear to tell the full story. Other research suggests that the effects and impact of caretaking are not always negative for the caregiver. It has been suggested that the subjective perceptions of the caregiver, unique for each person, play a critical role in their perceived burden, anxiety, and depression [13,20,21,22]. While it appears that when caregiving demands exceed psychological or social resources to cope, the result can prove detrimental to the caretaker’s emotional and physical health; yet, it has been suggested that this stress process model should also include the healthy caregiver hypothesis [23]. This model contends that healthier people are more likely to become caregivers and to remain in caregiving roles over time, and may actually experience health benefits from the prosocial behaviors that accompany this type of role [24]. Therefore, perhaps theoretical models on caregiver burden that focus exclusively on the negative impact of caregiving and suggest that caregivers, as a whole, are more stressed than non-caregivers, are missing critical pieces of the picture.

Several studies, in fact, have found that many family caregivers report little, or no, strain associated with providing caregiving assistance. Schulz and Beach, for example, found that 44% of the spouse caregivers reported “no strain” in association with caregiving tasks [19], and similarly, Roth [25] found that 33% of caregivers reported “no strain” and only 17% reported “a lot of strain.” Furthermore, a recent survey by the National Opinion Research Center [26] found that 83% of caregivers viewed it as being a positive experience. For example, caregivers may appreciate the positive experiences that come from sharing the limited time remaining with their family member [27,28]. In fact, the positive experiences of caregiving could potentially buffer against some of the possible stress-related health consequences.

One reason for these mixed results may have to do with caretakers’ level of life satisfaction prior to their new role. While some studies [1,29] have suggested that life satisfaction is influenced by the perceived burden of caregiving (meaning caregivers with a low degree of burden experience higher satisfaction), other studies [30,31] suggest that perceived burden of caregiving is instead influenced by low life satisfaction [32].

Furthermore, as noted by Lazarus [33], stress is subjective and can be defined as a “particular relationship between the person and the environment that is appraised by the person as taxing or exceeding his or her resources and endangering well-being”. Using this definition, people may not view caretaking as stressful if they are confident they have sufficient resources and feel the role is within their scope of knowledge [2].

Indeed, it would appear that many caregivers experience both rewards and strains simultaneously [34,35,36]. They may experience both emotional distress and psychological satisfaction and growth, effects that are not on opposite ends of the same continuum. Despite this growing amount of research pertaining to caregiving for human family members, little attention has been placed on caregiving for companion animals.

### 1.2. Caregivers and Companion Animals

Companion animals are increasingly popular, with 70% of U.S. households, or about 90.5 million families, owning a pet, and 38.4% owning at least one dog [36]. Typically seen as more than “just a dog”, 85% of owners consider their dog to be a member of the family [37]. Numerous studies have demonstrated the benefits of pets [38,39,40]. Pet owners, compared with non-owners, are more physically fit [41,42,43,44], have lower levels of depression [45], higher social functioning [46] and enhanced social support [44,45,47,48,49].

Yet, because dogs age more rapidly than people and have a shorter lifespan, acquiring a dog often includes ultimately caring for a senior dog (the exact definition of senior varies, dependent on several factors including breed and size) [50]. A dog’s life can be divided into four stages, including pediatric, adult, senior (mature, middle age), and traditional geriatric (senior/super senior). The senior/middle-age years typically include the transition period between healthy adult years and the traditional geriatric stage in which serious age-related issues are more common [51,52].

As a result of recent advances in veterinary medicine, including senior diets, improved dental care, pain management plans, and new drugs, many animals are living longer [53]. A longer life span, however, means that more dogs are faced with chronic ailments like arthritis, cardiovascular problems, cognitive dysfunction, and sensory impairment [54]. Caring for an elderly dog can be physically and emotionally challenging, yet the impact on owners’ lives when caring for an aging dog has received minimal scientific attention [55].

The handful of studies that have been conducted on caring for an aging or sick dog suggest that many aspects mirror those witnessed in human caretaking. For example, some owners talk about necessary changes in their daily routines, including feeling an obligation to leave their dog alone as little as possible [56]. Others report feeling depressed, guilty, and sorry for their pet [57]. Yet other owners indicate that caring for their aging dog does not negatively impact their own quality of life [58,59], with some owners noting they appreciate the extra time they are able to enjoy with their pet [60].

One qualitative study that investigated the impact of caring for an aging dog noted that most owners referred to their dog as a child or family member and, similar to the commitment they would feel in caring for a human family members, felt a strong obligation to caring for their dog [55]. Many participants in this study felt that not providing care because it was inconvenient or challenging was simply not an option [55]. Furthermore, many owners found caring for their aging dog rewarding because of the ability to spend time together, and also because they were able to live up to their own perceptions of a good dog guardian [55].

These studies, in both human and companion animal literature, suggest that caring for an aging (senior) dog may offer rewards as well as challenges. This current study was designed to better understand dog guardians’ experiences and perceptions related to caring for their aging dog.

## 2. Materials and Methods

### 2.1. Survey Development

This study used an exploratory mixed method design that first started with an exploratory online survey followed by a concurrent nested mixed methods design (where the quantitative and qualitative data are collected at the same time but one data set is dominant) [61]. In this study the quantitative survey data was the dominant data with the qualitative data adding further understanding and breadth [62]. A preliminary qualitative survey was given via social media and word of mouth to help inform the measures used in the current study. The qualitative survey yielded 25 responses to three open-ended questions. The questions asked respondents to describe what aging means in terms of living with an aging dog as well as ways in which their dog’s aging has had a positive or negative impact on them and/or their dog in terms of their relationship, activities and/or experiences. Responses to these questions were used to ensure the quantitative survey included the aspects most often identified as related to owning an aging dog.

A thematic analysis was carried out for each question to identify the core themes that were then used to inform survey development. Thematic analysis is a qualitative research method that is used to describe and analyze consistent and competitive themes in human experience [63]. The transcripts were read to develop a general sense of the themes that arose from the data. All transcripts were reviewed and analyzed by all four authors to reduce bias and increase confirmability of the results by including multiple researcher perspectives. Any discrepancies in coding were discussed between the authors to ensure agreement and to refine the themes. Similar codes were initially clustered and then consolidated into larger themes to gain an understanding of the data.

When asked what aging means to them regarding their aging dog, half of the respondents’ comments related to physical changes (e.g., reduced mobility). Yet, for many, aging was viewed in a positive light where “old is gold”. Examples include those who mentioned their dog is less anxious than when younger, or that they cherish the beautiful memories they are making with their dog. One frequently mentioned positive sentiment was a strengthening of the human animal bond and feeling that their dog is a member of the family. In addition, guardians also reported enjoying shared activities together and the companionship of their older dog. The most common comments about the negative effects of aging related to the general slowing down or decreased physical abilities of their dog. Other negative changes included increased vocalization, physical changes, and their own anxiety about their dog’s vulnerability and aging process.

An anonymous, online Qualtrics (Qualtrics, Inc., Provo, UT, USA) survey was designed, reviewed, and tested by the co-investigators and distributed between August and September 2021. The themes from the qualitative survey were used to identify relevant scales in the literature (i.e., The Lexington Attachment to Pets Scale (LAPS) and Pre-Death Inventory of Complicated Grief-Caregiver Version (Pre-ICG)) and to develop scale items to capture both the negative and positive experiences of caregiving and perceived caretaker support. These scales are described in greater detail below.

### 2.2. Participants

The resultant survey was piloted by a small group of individuals for ambiguity and potentially missing response options with applicable revisions made based on their feedback. The final survey (see Appendix A) and study design were approved by the Colorado State University Institutional Review Board (IRB #2554, 30 July 2021). Survey respondents were recruited through Amazon’s Mechanical Turk (MTurk; Amazon Inc., Seattle, WA, USA) platform, an open online marketplace providing affordable access to potential survey respondents. Diversity of participants recruited through MTurk is higher than typical Internet samples or American college-based samples. The data from MTurk has been found to meet acceptable psychometric standards [64]. To control for in-attentive participants, bots, virtual private networks (VPNs), and multiple submissions [65], we included two attention questions, and utilized the resources that can be embedded into Qualtrics surveys.

Adult (18 years or older) participants who were the current guardians of at least one aging dog and had owned the dog for at least 3 years were recruited for the study. In order to minimize the influence of geographic and cultural differences on respondent data, the survey was made available only to guardians living in the United States.

### 2.3. Procedures

In addition to guardian demographics (e.g., age, gender, ethnicity, education level, employment status and workplace (at home, away from home, etc.) and living arrangement (live alone, with other adults and/or children), respondents were asked how long they had lived with their dog.

Participants were next asked to complete The Lexington Attachment to Pets Scale (LAPS) [66]. The LAPS is a widely used instrument to measure attachment of people to their pets and contains 23 items, scored on a Likert scale from 0 (strongly disagree) to 3 (strongly agree) with a possible range between 0 and 69.

Next, participants were given a series of questions about possible lifestyle changes due to their aging dog. They were then asked to indicate behavioral and physical changes in their dog due to aging, as well as a series of statements that might reflect their own emotional and behavioral responses regarding their dog’s aging. The next series of questions pertained to their perception of the support they have received in caring for their dog.

Participants were also given an adaptation of the Pre-Death Inventory of Complicated Grief-Caregiver Version (Pre-ICG). The pre-death version of the Inventory of Complicated Grief (ICG) assesses grief over the expected loss of a loved one. The pre-death ICG has demonstrated high levels of internal consistency among caregivers (Cronbach’s alpha = 0.90) [67,68]. The version used in the current study consisted of four questions used by Tomarken (2008) and reported to have adequate reliability (0.76). The questions were answered using a 5-point Likert scale with 1 = Never to 5 = Always.

The quantitative data were analyzed using SPSS (IBM, Armonk, NY, USA). Descriptive statistics were calculated to characterize guardians and household demographics. New scales were created to reflect the following aspects of living with an aging dog: negative caretaking aspects, positive aspects, worry and anxiety, and social support.

We performed two multiple linear regression analyses: one to determine predictors of positive caretaking aspects of living with an aging dog and one to predict negative caretaking aspects. Results of exploratory univariate analysis of variances were used to guide the selection of predictors for both multiple regression models. Significance level (α) was *p* = 0.05 and all tests were two-tailed.

Finally, participants were asked to share one story or example of something that stands out to them about living with an aging dog—either positive or negative. The rationale for including qualitative analysis was to help interpret and illustrate the results provided by the quantitative data. The stories and examples are used to help extend the quantitative results to help elucidate the paradoxical effect of caregiving—the potential to simultaneously be both a positive and negative experience for dog guardians. From the 285 completed surveys, we received 216 narratives about living with an aging dog that varied in length and details. First, the authors independently read through the quotes and selected those that they felt best captured the positive and negative experiences of caring for an aging dog. This reduced the 216 accounts to 53. Together, the authors reviewed the 53 comments and discussed their content and meaning. The authors coded the content of these selected statements to identify the specific nature of the caretakers’ experiences. Specific quotes and examples that offered the best representation of the positive and negative experiences were then selected.

## 3. Results

A total of 284 participants completed the survey. The sample consisted of 140 (49.3%) female, 138 (48.6%) male (*n* = 129), and 6 (2.1%) nonbinary/other participants. The sample was 75.7% White, 8.5% African American/Black, 7.0% LatinX/Hispanic, 4.2% Asian, 2.1% Multi racial/multiethnic, 0.7% American Indian/Native Alaskan, 0.4% Middle eastern/north African (MENA), and 1.4% other or prefer to not say. The age of participants ranged from 18–29 years of age (133, 46.8%), 30–39 years (82, 28.9%), 40–49 years (43, 15.1%), and 50 and older (26, 9.2%).

When asked about employment status, the majority were employed full time (203, 71.5%), followed by unemployed (30, 10.6%), employed part time (27, 9.5%), retired (5, 1.8%), furloughed (3, 1.1%), other (12, 4.2%), and prefer to not say (4, 1.4%). For those who reported working (*n* = 230), 97 (42.2%) reported working at home and away from home, 65 (28.3%) reported working mostly/all the time at home, and 63 (27.4%) reported working mostly/all the time away from home. Five (2.2%) people indicated they preferred to not say. It should be noted that the survey was completed during COVID-19, such that more people than usual may be not working or working from home. Most participants lived with other adults (116, 40.8%) or other adults and children under the age of 18 (78, 27.5%). Fewer participants reported living only with children (48, 16.9%) or alone (31, 10.9%). The majority of participants reported having a university degree (146, 51.4%) or some college (67, 23.6%), with fewer reporting having a higher degree (52, 18.3%), high school/GED (18, 6.3%) or prefer to not say (1, 0.4%). When asked how long they had lived with their current dog, 122 (43.0%) reported 3–5 years, 71 (25.0%) reported 5–7 years, and 91 (32.0%) reported more than 7 years.

### 3.1. The Lexington Attachment to Pets Scale (LAPS)

Possible scores for the LAPS were between 0–69. In this study, the mean was 55.0 (SD 9.37), with a minimum of 20 and a maximum of 69. Cronbach’s alpha was 0.84.

### 3.2. Pre-Death Inventory of Complicated Grief-Caregiver Version (Pre-ICG)

The mean score for the four questions of the Pre-Death Inventory of Complicated Grief-Caregiver Version (Pre-ICG) was 2.95 (SD 1.14), with a Cronbach’s alpha of 0.88.

### 3.3. Negative Aspects of Caretaking Scale

A new scale was created to depict the negative aspects of caretaking for an aging dog. The items in the new scale included four items created for this survey and the four Pre-ICG questions (Table 1). The Cronbach’s alpha for this scale was 0.86.

### 3.4. Worry and Anxiety Scale

Seven survey items were combined to create the Worry and Anxiety Scale, with a resultant Cronbach’s alpha of 0.69 (Table 2).

### 3.5. Positive Aspects of Caretaking Scale

The scale for Positive Aspects of Caretaking was created by summing seven survey items, with a Cronbach’s alpha of 0.72 (Table 3).

### 3.6. Caretaker Support Scale

The Caretaker Support Scale was created by summing six items, with a resultant Cronbach’s alpha of 0.72 (Table 4).

### 3.7. Quantitative Results: Negative Aspects of Caretaking Scale

A Univariate Analysis of Variance test was performed to explore the relationship between negative aspects of caretaking (measured with the Negative Aspects of Caretaking Scale) and dog caretaker characteristics (gender, age, workplace) and the LAPS, Worry and Anxiety Scale, Positive Aspects of Caretaking Scale, and Caretaker Support Scale. The factors that were significantly associated with the Negative Aspects of Caretaking Scale included workplace, LAPS, Worry and Anxiety Scale, Positive Aspects of Caretaking Scale, and Caretaker Support Scale (Table 5).

Based on these results, multiple linear regression was conducted using the significant factors to determine their relationship with negative aspects of caretaking for aging dogs. The multiple regression model (Table 6) predicting the Negative Aspects of Caretaking Scale using LAPS, Worry and Anxiety Scale, Positive Aspects of Caretaking Scale, and Caretaker Support Scale, and workplace was significant (F5 = 35.18, *p* < 0.001), with an *R*^2^ of 0.445. Significant predictors of the Negative Aspects of Caretaking Scale included LAPS (B = −0.185; *p* < 0.001), Worry and Anxiety Scale (B = 0.688; *p* < 0.001), Positive Aspects of Caretaking Scale (B = 0.743, *p* < 0.001), and Caretaker Support Scale (B= −0.426; p = 0.011). The largest predictors of negative aspects of caretaking were the Worry and Anxiety Scale and the Positive Aspects of Caretaking Scale.

### 3.8. Quantitative Results: Positive Aspects of Caretaking Scale

A Univariate Analysis of Variance test was also performed to explore the relationship between positive aspects of caretaking (measured with the Positive Aspects of Caretaking Scale) and guardian characteristics (gender, age, workplace), and the LAPS, Worry and Anxiety Scale, Negative Aspects of Caretaking Scale, and Caretaker Support Scale. The factors that were significantly associated with the Positive Aspects of Caretaking Scale included LAPS, Negative Aspects of Caretaking Scale, and Caretaker Support Scale (Table 7).

Multiple linear regression was conducted using the significant factors from the Univariate Analysis of Variance test to determine their impact on positive aspects of caretaking. The multiple regression model (Table 8) predicting the Positive Aspects of Caretaking Scale using LAPS, Negative Aspects of Caretaking Scale, and Caretaker Support Scale was significant (F3 = 65.11, *p* < 0.001), with an *R*^2^ of 0.411. Significant predictors of positive caretaking included LAPS (B = 0.203; *p* < 0.001), Caretaker Support Scale (B = 0.207; *p* = 0.016), and Negative Aspects of Caretaking Scale (B = 0.279, *p* < 0.001). The largest predictor of the Positive Aspects of Caretaking Scale was the Negative Aspects of Caretaking Scale.

### 3.9. Qualitative Results: Compasionate Care as a Double-Edged Sword

One of our participants wrote: “Living with an aging dog makes me think about my companionship with animals much more, it also makes me think about how I am going to deal with loss and grief when they eventually pass away, and it makes me sad.”

Caring for others, whether human or animal, can simultaneously be satisfying and fulfilling, as well as upsetting and overwhelming (Beach et al., 2000, Brønden et al., 2003, Christiansen et al., 2013, Harmell et al., 2011, Lawton et al., 1991). This premise is supported by our regression results, whereby the positive and negative aspects of caretaking are positively related to one another (Table 6 and Table 8). The qualitative stories and examples provided by our study participants are helpful in clarifying the complex nature of caring for an aging dog. First, we share some of the reported negative experiences of living with an aging dog including both physical and mental challenges. We then report on some of the positive changes (e.g., calmer, quieter, a need for less physical exercise, etc.) associated with an aging dog. Finally, we discuss how the bond between aging dogs and their caretakers creates a unique bittersweet moment in time of deepening affection and appreciation, coupled with anticipatory grief and sadness.

As expected, participants described many different challenges of caring for an aging dog, as well as the anticipated painfulness of their dog’s death. As one participant commented: “*As my dog ages, I’ve noticed more and more little health issues start to crop up and it makes me become painfully aware that the number of days I have left with my dog is dwindling. My dog has been with me through most of my major life milestones so far, and seeing his health deteriorate has taken a bigger mental toll on me than I’ve ever expected. My dog takes a long time to get out of bed and I worry every day that one day he will not wake up*.” In addition, participants identified the physical changes in their aging companion such as slowing down because of arthritis, and stiffness or pain resulting in less energy and stamina for going up stairs, playing, and going for walks. For example: “*I first really noticed that my dog was aging when I was going to take him on a walk and went to grab the leash. He usually runs up behind me, extremely excited to go for a walk. However, I looked back and realized he was having trouble standing up. It was a very sad moment for me. His legs began rapidly developing arthritis, and it became more difficult to get around*.” Another noted: “*Sometimes, when we go out for a walk, she gets tired a lot faster than she used to, and I have to pick her up and carry her home. It breaks my heart a little bit that she has some trouble moving on her own now, but that’s a part of life*.” Other common physical challenges mentioned include managing more health concerns, changes in the dog’s appetite or diet, and the need to go outside more often or having more accidents in the house.

Participants also described the challenges associated with their dog losing their hearing or sight, which was often linked to their dog becoming more anxious or easily confused or frightened. For example, “*As he’s grown older, my dog has become increasingly deaf. If I’m not careful, I can walk up behind him and startle him. He jumps and seems genuinely frightened until he sees who it is. It’s a reminder that I need to adapt to take care of his needs*.” Additionally, participants described emotional changes including greater separation anxiety or confusion. One person noted “*Something that has really affected us as our senior dog ages is his episodes of canine cognitive dysfunction at night. He often wakes up extremely restless, anxious, and confused and not only does it affect my sleep, it is very upsetting to see and not be able to help much.*”

In addition to the negative aspects of living with an aging dog, participants also described some of the more positive changes. As one person noted: “*Aging dogs are always like an elder family member so matured and calm, always understanding our emotions and feelings. We are so blessed to have such caring dogs around us.*” Another commented: “*She has become much more gentle with her interaction with people. She doesn’t jump up as much when greeting new visitors, and gives a soft paw when prompted. She has become much more affectionate*.” Older dogs were often described as calmer, as well as more mature, relaxed, cuddly/affectionate, tolerant of strangers or other dogs, and attentive to their guardian’s emotions. One participant noted: “*My dog is more mature, shows more affection and likes to relax.*” Another stated: “*As much as I get sad thinking about the day she is no longer with me, I love the new sense of calmness that has come with her age.*” Participants also described how their older dog gets along better with other animals, dogs and cats: “*He’s just perfect and sweet! He’s a lot more relaxed around the other animals, even when they play a little too hard. He’s an angel*.”

The participants’ comments also illustrate how the strength of the human-animal bond is vital to their willingness and patience in providing compassionate care to their dog, where their compassion reflects an awareness of the suffering of another and the desire to ease it. A strong human-animal bond, however, also makes observing the pain and suffering of their aging dog, and their inability to alleviate their suffering and the inevitability of their death, an upsetting experience. For example, one participant’s comments illustrate the conflicting feelings of joy and sadness that are experienced simultaneously: “*I would say one thing that stands out is that you really understand how much of a gift to the dog in your life is, and that the sadness over their aging is easily trumped by the joy the dog brings you*.” Many participants described how their bond with their dog strengthens and they feel more emotional attachment and affection as their dog ages: “*Something that stands out to me the most are our times spent together. They’re more affectionate and seem to be more meaningful, like I know that moments like these are becoming limited, so each one becomes more and more special. My dog also isn’t the most cuddly, so for her to come lay by my side, those are the little moments that I love.*”

Another interesting theme from participants’ comments is how they explicitly refer to the fact that they do not resent their aging dog’s need for additional care. They often acknowledge the reciprocal relationship of the bond their share with their dog in where their younger dog was faithful, “there for them” and now it’s their turn to be there for their dog. For example, “*It’s bittersweet and sad, yet I will gladly take care of my aging dog because he gave me the best years of his life and I want to repay him by being there for him when he needs me, like he was always there for me when I needed him*.” Another suggests how the bond with their aging dog hasn’t diminished but strengthened as their dog requires more care: “*I don’t consider myself an exceptionally patient person, but with him I do what I need to and feel no resentment. His old age has made me appreciate him so much more, and I rarely feel upset being a witness to it. I’m happy to have grown up with him and watch him grow old*.”

Participants also described significant changes and sacrifices that they have made to their lifestyle in order to provide extra care for their dog. Some examples include shorter or fewer walks, carrying their dog up the stairs or into the car, purchasing a wagon for the dog when it’s too tired to walk any further, getting up more often at night to let the dog outside, forgiving accidents in the house, letting the dog sleep on the bed, and cooking special meals that are easier to digest. Several participants mentioned that working from home allows them to offer better care for their dog: “*I never really worried about my dog aging until he started having stomach issues and occasional incontinence. This one thing changed my lifestyle quite a bit—I started working from home and staying home as much as possible just in case. But I don’t resent him for it at all; I feel blessed that I have the kind of job where I can be here for him when he needs me.*” Another wrote “*I heated my pool so I could swim with my 14-year-old choc lab. I worked from home, so we swam daily*.” These lifestyle changes and sacrifices reflect the genuine compassion and unwavering sense of duty that many animal guardians feel for caring for their dog as it ages. For dog guardians, caring for their aging dog, despite the inevitable outcome, can be both fulfilling and rewarding.

## 4. Discussion

Sharing one’s home with a companion dog offers a multitude of both physical and psychological benefits [38,39,40,43,46,49], but due to a dog’s relatively short lifespan, also typically includes aging and ultimately, loss. The impact on guardians when caring for an aging dog appears to share many similarities with caregivers of human family members. These changes often include practical, pragmatic ones such as altering one’s lifestyle or daily routines to ensure they are able to care for their loved one [1,56,69]. As noted by Christiansen et al. [55], many dog guardians deal with these changes by accepting the fact that caring for their dog includes changes that can be time consuming, burdensome, and inconvenient.

In addition to logistical changes, similarities between human and canine caregiving can be seen in the emotional impact of caretaking. Decades of research pertaining to human caregiving has found that many caregivers struggle with stress, anxiety and depression [3,18,70], often mitigated by numerous factors, including mutuality (the positive qualities of the relationship), perceived support and available resources [1,30,71]. Many pet guardians also report feeling depressed and burdened [55,57,72,73]. Yet, others report more positive feelings associated with a severely ill companion animal [54,60]. Our study found that the impact of caring for an aging dog is in fact a complex interwoven myriad of feelings that often include both positive and negative emotions and experiences.

For example, we found that guardians who reported more negative thoughts and ruminations about their dog’s aging were more likely to feel worried and anxious. The stories shared by participants highlight this quantitative finding and illustrate how painful it can be for some guardians to witness their dog slowing down and facing increasing physical and mental health-related challenges. Additionally, many guardians struggle with feelings of anticipatory loss and grief, defined as the fear of losing a significant other [74]. This anticipatory loss, coupled with worry and anxiety, are similar to that reported by caretakers of human family members [13]. A unique factor for aging pets that can add to the stress associated with the aging process is the option for euthanasia. The decision to euthanatize, including an ongoing assessment of their dog’s quality of life, adds an additional element of stress and worry for many dog guardians. Euthanasia decisions involve complex issues, none of which are typically black or white. Trying to discern their dog’s quality of life, assessing the impact of medical interventions, and trying to assess what is in the dog’s best interest, can be overwhelming and exhausting.

Yet, it is important to note that not all caregivers experience worry, anxiety and depression. Instead, it appears that these emotions are strongly correlated with subjective caregiver burden in caring for both humans and companion animals [13,20,21,22,75]. This perception of burden is related to numerous factors including whether the demands exceed the guardian’s psychological or social resources to cope, feelings of mutuality, and their own mental and physical health [19,25,71,72,76]. So, while for some, the role of family caregiver is overwhelmingly burdensome and has a negative impact on emotional and physical health [5], these responses are not universal and do not preclude more positive feelings.

Another factor that impacts feelings associated with caregiving is the bond guardians share with their dog. Our results suggest that having a stronger emotional attachment to one’s aging dog reduces caretakers’ negative feelings (e.g., feelings of guilt, resentment, longing for when their dog was younger). We learned from our participants’ stories that many do not resent their aging dog’s additional care and they gave a variety of examples of the different sacrifices they willingly make to provide extra care. Both the quantitative and qualitative results also highlight the positive experiences and emotions associated with living with an aging dog (e.g., sense of purpose, caring and cherishing their dog). Not surprisingly, stronger emotional attachment and feeling support from others both predict a more positive experience for animal caregivers. Many participants described how they feel closer to their aging dog and find them to be more affectionate, cuddly, quieter, and mature as they age. Similar positive sentiments regarding the benefits of a caretaking role have been reported by human caretakers [26,27,28].

Furthermore, not only can caregivers experience both positive and negative emotional reactions to their caregiving role, our results suggest that the two are correlated. That is, the more animal caretakers feel a sense of purpose and cherish their time with their aging dog, the more guilt and resentment they appear to feel in caring for their dog and the more they long for the time when their dog was younger. In addition, the quantitative results also revealed that a stronger emotional attachment to their dog predicted increased negative and positive emotions related to living with an aging dog. We learned from the qualitative findings the impact of the human-animal bond in shaping caretakers’ bittersweet experiences of watching their long-time companion slowing down; feelings of sadness, while also cherishing the time they spend caring for them.

There are several limitations to this study that should be noted. First, the results are based on a small sample of self-selected individuals answering a survey through Amazon Turk. Although our results contribute to our understanding of the complex nature of care giving to aging dogs, caution should be taken when generalizing to the entire population of dog guardians in the U.S. Additionally, the survey was conducted during the COVID-19 pandemic, and we do not know if the pandemic influenced the nature of the relationship between guardians and their aging dogs. Furthermore, the survey did not explore the degree of age-related changes in the participants’ dogs, only that the participants recognized changes in their dog’s behavior that they attributed to age. As such, we do not know the role that certain common disease states, such as osteoarthritis, cancer, cognitive dysfunction, and diabetes had in guardians’ perceptions of their dogs’ aging process. Finally, we did not collect any medical data on the dogs to determine overall health status and how that may have affected the relationship with their guardian. Future research exploring the effect of both the guardian’s and dog’s health and resources available to care for the dog on the relationship between guardians and their aging dogs would be of value. Additionally, expanding this line of inquiry to the guardians of other types of companion animals can help to determine if the relationship found in this study is unique to humans and dogs, or if it can be applied to other types of companion animals too.

## 5. Conclusions

In most cases people will outlive their dogs, resulting in the need to care for an aging dog. The fact that a deep bond with an aging dog can increase both positive and negative feelings is vitally important in understanding the caregiving role. Our findings suggest that these feelings are not opposites on the same continuum, but instead, they correlate, whereby increased feelings of satisfaction, a sense of purpose and moments of contentment and happiness are often accompanied by increased worry and concern, often within the context of anticipatory grief.

This knowledge can be used to help support companion dog caretakers. For example, helping guardians identify the positive aspects of their new role may help them focus on the benefits that can accompany caring for an aging dog. As noted by many of the participants, these benefits include positive feelings associated with being able to give back to, and provide for, their steadfast companion. The provision of support for caregiving guardians should also include addressing potential anticipatory grief and helping them identify strategies that can help in mitigating their anxiety. This might entail exploring ways to implement changes in their daily routines to enhance their dog’s quality of life (e.g., special treats, allowed on the furniture, etc.) or permit more time together. It may also involve the process of initiating thought and discussion about how they may want to create a lasting bond with their dog when he/she is no longer physically present. Creating videos or photo albums or other ways to memorialize their dog may be of value. Above all, it is important to recognize that the experience of caring for an aging dog is individualized and it should not be assumed that the caregiving role is filled only with hardship and pain. Instead, it would appear that for many guardians, caring for an aging dog is a complex dynamic with both positive and negative factors that offers an opportunity to deepen their bond and create positive, rewarding experiences and memories.

## Figures and Tables

**Table 1 animals-12-01074-t001:** Survey items that constitute the Negative Aspects of Caretaking Scale.

I would like to be able to walk/run further with my dog than he/she can now walk
I feel guilty when I exercise or go for a walk and can no longer take my dog with me
There are times I resent the changes I have had to make in my daily schedule to care for my dog
I dread leaving my dog for any period because of his/her age
* I feel myself longing and yearning for my dog as he/she was before aging
* I feel that life is empty and meaningless without my dog being healthy
* I am bitter over my dog’s aging
* I think about my dog’s aging so much that it can be hard for me to concentrate on anything else or do the things I normally do

* Pre-ICG question.

**Table 2 animals-12-01074-t002:** Survey items that constitute the Worry/Anxiety Scale.

I worry how the loss of my aging dog will affect me and my family
I worry that the number of remaining days with my dog are limited
I worry a great deal about when my dog can no longer get around by him/herself
I am worried other dogs will accidently hurt my aging dog
I worry about my ability to afford veterinary care for my aging dog
My dog gives my life purpose, and I am worried about what I will do without him/her
I dread the day my dog is no longer with me

**Table 3 animals-12-01074-t003:** Survey items that constitute the Positive Aspects of Caretaking Scale.

Caring for my aging dog gives me a sense of purpose
I tend to bend my dog-related rules more as my dog ages (i.e., I let my dog sleep on the couch or bed, I give treats more often)
I find I am more protective of my dog as he/she ages
I cherish the time I spend with my aging dog
The amount of time I spend with my dog
His/her ability to understand your feelings and know what you are thinking
How affectionate he/she is

**Table 4 animals-12-01074-t004:** Survey items that constitute the Caretaker Support Scale.

I talk with friends or my family about my concerns related to my aging dog
I have talked to my vet about my concerns related to my aging dog
I feel my vet and I are a team when it comes to caring for my aging dog
I wish I had someone to talk to about my aging dog *
How much I socialize
I feel my family and/or friends do not understand what is needed to care for an aging dog *

* reverse coded.

**Table 5 animals-12-01074-t005:** Univariate Analysis of Variance test results assessing the association between the Negative Aspects of Caretaking Scale and LAPS, Worry and Anxiety Scale, Positive Aspects of Caretaking Scale, Caretaker Support Scale and workplace.

ANOVA					
Model	Sum of Squares	df	Mean Squares	F	Sig.
**Worry/anxiety**	**1428.88**	**1**	**1428.88**	**47.34**	**<0.001**
**Positive aspects**	**1566.03**	**1**	**1566.03**	**51.88**	**<0.001**
**Caregiver Support**	**160.96**	**1**	**160.96**	**5.33**	**=0.022**
**LAPS**	**351.89**	**1**	**351.89**	**11.66**	**=0.001**
Gender	7.57	1	7.57	0.25	=0.617
Age	88.41	3	29.47	0.98	=0.405
**Workplace**	**538.61**	**2**	**269.30**	**8.92**	**<0.001**

Bold denotes significance.

**Table 6 animals-12-01074-t006:** Results of the multiple linear regression model predicting the Negative Aspects of Caretaking Scale.

ANOVA						
Model	Sum of Squares	df	Mean Squares	F	Sig.	
Regression Residual Total	5720.97 7123.19 12844.16	5 219 224	1144.19 32.53	**35.18**	<0.001	
**Coefficients** Dependent Variable: the Negative Aspects of Caretaking Scale)						
Variable			Coefficient (B)	Std. Error	t	Sig.
(Constant)			1.25	4.14	0.30	=0.763
**LAPS**			**−0.19**	**0.05**	**−3.96**	**<0.001**
**Worry/anxiety**			**0.69**	**0.10**	**6.68**	**<0.001**
**Positive aspects**			**0.74**	**0.10**	**7.62**	**<0.001**
**Caregiver Support**			**−0.43**	**0.17**	**−2.58**	**=0.011**
Workplace			−0.94	0.52	−1.79	=0.074

Bold denotes significance.

**Table 7 animals-12-01074-t007:** Univariate Analysis of Variance test results assessing the association between Positive Aspects of Caretaking Scale and LAPS, Negative Aspects of Caretaking Scale, and Caretaking Support Scale.

ANOVA					
Model	Sum of Squares	df	Mean Squares	F	Sig.
Worry/anxiety	58.75	1	24.30	1.96	=0.163
**Caregiver Support**	** 372.39 **	** 1 **	** 58.75 **	** 4.74 **	** =0.031 **
**LAPS**	** 643.22 **	** 1 **	** 372.39 **	** 30.04 **	** <0.001 **
**Negative aspects**	** 1.40 **	** 1 **	** 643.22 **	** 51.88 **	** <0.001 **
Gender	28.04	1	1.40	0.11	=0.737
Age	4.25	3	9.35	0.75	=0.521
Workplace	58.75	2	2.13	0.17	=0.843

Bold denotes significance.

**Table 8 animals-12-01074-t008:** Results of the multiple linear regression model predicting the Positive Aspects of Caretaking Scale.

ANOVA						
Model	Sum of Squares	df	Mean Squares	F	Sig.	
Regression Residual Total	2553.51 3660.33 6213.84	3 280 283	851.17 13.07	65.11	<0.001	
**Coefficients** (Dependent Variable: Positive Aspects of Caretaking Scale)						
Variable			Coefficient (B)	Std. Error	t	Sig.
(Constant)			16.45	1.94	8.49	**<0.001**
**LAPS**			**0.203**	**0.023**	**8.65**	**<0.001**
**Caregiver Support**			**0.207**	**0.086**	**2.41**	**=0.016**
**Negative Aspects**			**0.279**	**0.028**	**9.80**	**<0.001**

Bold denotes significance.

## Data Availability

Data available upon request from corresponding author.

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
