# Peer review of "Canine Caregivers: Paradoxical Challenges and Rewards"

_animals, 2022, doi:10.3390/ani12091074_

Round 1

Reviewer 1 Report

This is a very interesting study that provides new insights into the experiences of people providing care for older dogs. 

One important aspect of the caregiving experience that was not directly addressed in the manuscript was mutuality, the quality of the relationship between the caregiver and one receiving care. There is a great deal of evidence in the human caregiving literature about the impact of this relationship on caregiving outcomes, including burden and satisfaction. The results of this study illustrate how the pre-existing relationship with the one receiving care (i.e., a pet dog) shapes the caregiving experience. The link between mutuality and the human-animal bond may be worth exploring in the discussion section of the paper.   

The 'Materials and Methods' section would benefit from subheadings (e.g., 'development of the survey', 'participants', 'procedures'). 

Please provide more detail on the thematic analysis employed to identify the core themes used in survey development (e.g., how many people were involved in this analysis) (lines 160 to 166).

Likewise, more information is needed on the analysis of the open-ended item at the end of the survey (lines 227 to 236).

 The aim of the study was to understand experiences rather than develop a survey. Therefore, the methodology in the survey development is appropriate. However, at least one scale has been developed to measure caregiver burden in pet owners (Spitznagel, M. B., Mueller, M. K., Fraychak, T., Hoffman, A. M., & Carlson, M. D. [2019]. Validation of an abbreviated instrument to assess veterinary client caregiver burden. Journal of veterinary internal medicine33(3), 1251-1259). This may present a limitation to the study.

In Table 7, is 'Positive Aspects' meant to be 'Negative Aspects'?

The implications and conclusions accurately reflect the results of the analysis. The findings provide important insights into the experiences of living with and caring for an older dog.   

Author Response

Thank you for the constructive review. Please see attached for our responses.

Reviewer 2 Report

Thank you for a very interesting and timely paper, which seeks to explore in more depth the ‘caregiver burden’ that has been applied to those caring for aging animals.

I have just a couple of queries /comments.

Lines 88-92 This sentence is a little difficult to unpick – perhaps it could be rewritten for clarity? I am still unsure what the ‘reverse of this relationship’ means.

In your methods section, you describe a concurrent nested mixed methods design, but go on to explain that you used an exploratory qualitative survey to decide on the scales used for the mainly quantitative survey. I would perhaps describe this as an initial exploratory mixed methods design, leading to a concurrent nested study.

How were the qualitative data from the quantitative survey analysed? I assume that thematic analysis was used, as this was the method of analysis for the initial qualitative survey, but I think this needs to be clarified.

In table 3, is the survey item ‘The amount of time I spend time with my dog’ correct?

Line 508, the phrase ‘a self a sense of purpose’ does not make sense.

Finally, reference 56 is incomplete.

Author Response

(The authors gave the same response as above.)

Reviewer 3 Report

The article is well written and the methodology is sound. My thoughts are that it is somewhat superficial, but provides a good basis for replication of the study addressing more variables. I think it is easy to say that we love our pets (and a possible limitation of the study) and understand that we need to care for them, especially in their senior years. What the study does not consider is things like: if there is more than one person living in the home, do they all contribute to the care; what is the income level of the caregiver as having the means for veterinary care, surgery, modified diet, etc. matters; are there other pets in the home and more. Personally, I think this gets us more into the reality of care giving for senior pets. 

I think the study as-is is suitable for publication and I would love to see a deeper dive into additional variables. Thank you for your hard work and contribution to the field.

Author Response

Thank you for the positive feedback. We agree, there are so many exciting ways to continue this line of research, including all of the variables you mentioned.